# Battery-Free and Real-Time Wireless Sensor System on Marine Propulsion Shaft Using a Wireless Power Transfer Module

**DOI:** 10.3390/s23020558

**Published:** 2023-01-04

**Authors:** Young Chul Lee, Van Ai Hoang

**Affiliations:** Department of Marine Electronic, Communication and Computer Engineering, Mokpo National Maritime University, Mokpo 58628, Republic of Korea

**Keywords:** wireless sensor system, propulsion shaft, wireless power transfer (WPT)

## Abstract

In this paper, we present a wireless sensor system (WSS) that integrated an inductive wireless power transfer (I-WPT) module for battery-free real-time monitoring of the status of rotating shaft in ships. Firstly, an optimized I-WPT module for seamless power supply was implemented using multiple Tx and Rx coils, and its power capability of 1.75 W with an efficiency of 75% was achieved. Secondly, as a result of the high-power transfer performance of the implemented I-WPT module, an entire WSS that integrated four sensors was designed on the rotary shaft. Finally, the designed WSS was installed on a small-scale test bench system with a shaft diameter of 200 mm; it was demonstrated that the status of a propulsion shaft could be monitored in real time without a battery.

## 1. Introduction

With the beginning of the fourth industrial revolution, digitalization is accelerating in all industries. The maritime industry is also once again at the forefront of the new era—one driven by increased digitalization and innovation, in particular, automated ships [1].

Automated ships, or smart ships, are the result of several sustainable innovations in traditional ships. However, there are still many technical challenges to be solved, and many new innovative policies and technologies are being proposed and studied in many institutes around the world [1]. 

One of the important key technologies for ship automation is the complete implementation of a real-time remote monitoring system, or a real-time wireless sensor system (WSS). In order to monitor the engine and propulsion shaft, as well as most of the equipment in real time in the engine room, their statuses must be precisely measured and reported to the management system, or to the remote control room. In particular, in the case of a propulsion shaft, various parameters such as torque, thrust, rotational speed, and shaft power must be measured and monitored in real time for safe navigation and maintenance. Currently, in most large ships, several sensing systems are powered by wire, and their sensing signals are also transmitted to the management system by wire. Wired-based sensor systems have limitations in installation time, cost, and scalability. 

Various attempts have been made to apply battery-free real-time wireless sensor sys-tems to a rotating shaft. These efforts can be largely divided into research on sensors [2,3,4] such as non-powered sensors, and research on power sources [5,6,7,8,9,10,11,12,13] such as wireless power transfer (WPT) or energy harvesters. For sensor research, powerless and non-contact sensor technologies are being considered for rotating shaft applications. Sensors that use an electrostatic charge film [2] or magnetostrictive patch [3] on a rotating shaft have been proposed, and these detect changes in induced current or electromotive force due to shaft vibration at a fixed electrode away from the shaft. However, since these technologies are affected by ambient noise, such as temperature change or electromagnetic fields, additional devices or circuits are required. Two-dimensional (2-D) position-sensitive detection (PSD) has been used for vibrational harmonics measurements, but complex calibrations such as additional sensors and spectrum analysis are involved [4]. Until recently, interesting sensor technologies have been devised for continuous real-time measurement, but additional devices and circuits are needed to achieve accurate and reliable measurements.

In the case of sensor systems that operate without a battery on the rotating shaft, energy harvesting or WPT technology can generally be considered as the power source. For energy harvester technology, it is still difficult to produce enough power to these drive sensor systems. In particular, the WPT technology commercialized in smart phones and home appliances has made outstanding technological progress. A variety of commercially available WPT modules and components are on the market, and have been used in a variety of applications because of their easy implementation [5]. Several configurations that use commercial WPT coils and modules for the shaft of a wind turbine were proposed and evaluated experimentally for driving an acceleration sensor and Bluetooth radio on the shaft. Their obtained power ranged from 0.1 to 1.2 W. Especially, co-axial configurations of the coils, where the coil is wrapped around the shaft so that the shaft is at the center of the coil, generate severe losses, such as eddy current and hysteresis losses [5]; moreover, the proposed configurations are difficult to apply to a large-scale shaft. A WPT system that uses magnetic coupling resonant coils with a diameter of 700 mm was reported for online telemetering system applications, and it achieved a maximum power transfer efficiency (PTE) of 83.7% [6]. However, improvement in the efficiency is limited due to the influence of the transmission shaft of the rolling mill, and additional circuitry to compensate for the parasitic capacitance of the resonant coils was suggested. A proposed WPT system using a U-shaped core and receiving coil on the PCB [7] verified its feasibility by achieving a PTE of 51.19% and an output power of 1.72 W. However, in order to be applied to the propulsion shaft of large marine engines, a more compact size is required. Rotary transformers [8,9,10] have been developed for WPT applications. However, they are mainly applied to the wireless excitation of electrically exited synchronous machines. Due to the complex structure and large size of rotary transformers, they are difficult to apply to the propulsion shaft of ships. A capacitive-WPT [11,12,13] technology using rotating capacitors has been reported, but it was mainly proposed as a replacement for slip rings. Since large-area electrode plates are used, the technology is also difficult to apply to a propulsion shaft with a diameter of 200 mm or more. An inductive WPT (I-WPT) module that uses multiple coils was reported for WSS applications on the propulsion shaft. By optimizing the air gap between transmitter and receiver coils and their number, a transferred output power of 1.75 W and PTE above 75% were achieved [14]. 

Currently, in the propulsion shaft of large ships, various situations such as vibration, rotational speed, and temperature change are reported to the main management system in real time. However, most of the sensors are individually mounted on the rotating shaft in a wire-based sensor system. Smart ships or autonomous ships need to integrate various sensors into one sensor system. Considering their ease of installation and scalability, WSSs that are capable of wireless-based power supply and communication are a key technology.

Therefore, in this study, we present a battery-free WSS that monitors the critical status of the propulsion shaft in real time. Sensors to detect vibration, rotational speed, and temperature change, as well as sensors for power monitoring, were integrated into a WSS. An I-WPT module with an optimized configuration that uses commercially available coils and modules is presented, and a WSS consisting of four sensors, a wireless communication module, and a control module, was designed in detail. Sensing of the shaft status and its wireless transmission were demonstrated, using only the power supplied by the WPT module.

## 2. Design of the Battery-Free and Real-Time Wireless Sensor System (WSS)

The propulsion shaft is a major machinery component on a ship propulsion system, and its operation should ensure efficiency, reliability, safety, and durability. Therefore, much attention has been focused on effective maintenance procedures. Condition-based maintenance is a scientific strategy that uses WSSs that decide the optimal maintenance times according to actual conditions of the machinery, in order to prevent unexpected faults, improve their reliability and availability, decrease downtime, and increase operating efficiency. 

As shown in Figure 1, the WSS and the multi-coil-based I-WPT unit [14] are attached to the surface of the rotating shaft. The I-WPT system supplies sufficient power to the WSS without replacing the battery, and since the wireless communication module including the antenna is integrated into the WSS, the signal measured by the sensor can be transmitted in real time. The whole system, except the fixture of the I-WPT unit, rotates with the shaft.

### 2.1. Inductive-Wireless Power Transfer (I-WPT)

WPT technology has already matured, and a variety of high-performance components are readily available in the market. The technology can be easily applied to various applications. However, in order to be applied to a rotating shaft, it must be designed so that power can be continuously transmitted without interruption while the shaft rotates. Therefore, it is necessary to optimize the configuration of the transmitter (Tx) and receiver (Rx) coils for propulsion shaft applications. 

Figure 2 presents the optimized configuration of the Tx and Rx coils on a large-scale shaft with a diameter of 200 mm [14]. Figure 2a shows a three-dimensional (3-D) view of the designed I-WPT module, and the equivalent circuit of the I-WPT system is shown in Figure 2b. The I-WPT module is composed of the Tx module, Tx coils, Rx module, and Rx coils. As a result of design optimization, the spacing between the Tx and Rx coils of 3 mm, the six Tx coils on the fixture, and the four Rx coils on the shaft were determined [14]. The magnetic fields generated by the Tx coils are transmitted to the Rx coils, which convert the fields back into electric power to provide power to the load through the Rx module. The Rx module supports voltage stabilization, and adds functionality as a rectifier. The fixture was used to support Tx coils fixed at a specified position on the inner face of its fixture. The commercial Tx and Rx coils and modules are shown in Figure 2c. The Tx coils have inner and outer diameters of 10 mm and 30 mm, respectively. The Rx coils have dimensions of 2.6 × 3.8 × 0.09 mm^3^. The diameter and number of turns in the Tx coils are 0.5 mm and 22, respectively. The Rx coils are aligned along the inner circumference of the fixture, and are located inside. The Tx coils are connected in parallel, and their internal resistance and inductance are 0.2 Ω and 10 µH, respectively. The four Rx coils are also connected in parallel, and are evenly distributed around the shaft. The resistance and inductance of the Rx coil are 0.18 Ω and 10.7 µH for 15 coil turns, respectively.

The fabricated I-WPT system was evaluated in terms of transmitted voltage, current, power, and efficiency. A power of 2299 mW (6.2 V, 372 mA) was input to the WPT Tx module, a 15 Ω load was connected to the WPT Rx module, and the voltage and current were measured at the output terminal of the Rx module. The measured output voltage and current were 5 V and 350 mA, respectively. A transfer efficiency of 75% was achieved [14].

### 2.2. Wireless Sensor System

Since a power of 1.75 W can be wirelessly transmitted to the rotating shaft using the proposed WPT module, a WSS (without a battery) that integrated all components, such as four sensors, a controller, an amplifier, and a radio module, could be designed on the rotating shaft. Figure 3 shows a block diagram of the battery-free and real-time WSS. It is composed of the WSS on the shaft and the management unit outside the shaft. In the management unit, I-WPT transmitter (Tx) module wirelessly supplies power to the WSS on the rotating shaft, and a current sensor monitors the amount of input power. The sensing signals generated on the shaft are wirelessly transmitted to the management unit for storage and analysis.

The WSS on the shaft consists of the power, sensing, control, and wireless communication part. The WSS can be used to measure the conditions of vibration, temperature, and speed of the rotating shaft, without the use of direct wiring. This WSS is wirelessly powered, and can be wirelessly controlled. 

The power partSince the power delivered from the I-WPT unit is not constant due to the changing shaft speed, it is necessary to use regulators to convert the power into stable power that is suitable for various sensors or modules. Two voltage regulators were used for this WSS; a 5.0 V power supply for the various sensors and control parts, and a 3.3 V power supply for the communication module. Two main features, such as over-current and over-temperature protection, were included.The sensing partFour sensors were installed on the shaft to measure current, speed, strain gauge, and temperature. Power monitoring is a basic function of the real-time WSS. In general, this can be achieved using the current sensor. Input power and power consumption should be monitored in real-time. The current sensor measures the input power transferred from the I-WPT unit, and monitors the power consumption of the WSS. The shaft speed is also one of the important parameters for a ship’s performance to monitor factors such as engine torque, sea state identification, fuel consumption, etc. The speed of the rotating shaft can be measured using a Hall-effect sensor, which detects the presence and magnitude of a magnetic field, and converts a varying magnetic field into an electric signal. In this study, the detector and permanent magnet were installed on the shaft and fixture, respectively. The torque generated by the main engine is transmitted to the propeller by virtue of marine propulsion shafting [15]. During real navigation situations, ships are usually perturbed by waves, wind, and many other exciting forces; all of these excitations will lead to vibration responses in the loading direction. Therefore, vibrations of the shaft should be measured in order to monitor the ship for damage diagnosis. The strain gauge sensors are used to measure the mechanical strain on the rotating shaft. A Wheatstone bridge-based sensing circuit was utilized, and it required an amplifier to boost measured low-level signals in order to feed them to an analog-to-digital converter (ADC). A temperature sensor was used to monitor the temperature variation of bearing cages during bearing operation, in order to monitor bearing health. The distance between the main engine and the propeller is relatively far, so the rotating shaft is commonly used. In this case, the bearings play an important role to reduce friction and allow for smoother rotation. Temperature change is one of the most important parameters that affects the functional life and performance of bearings. Rapidly changing temperatures ultimately lead to bearing failure. Therefore, temperature changes in the bearings must be monitored in real time.The control partThe Arduino control module had two main functions, which were as the data processor for the arrangement, calculation, storage, and display of sensing data, and as the central controller for the WSS, in order to control all sensors and modules on the shaft.The communication partThe communication part included a radio transmitter (Tx) and receiver (Rx) module. Detected data are transmitted wirelessly to a laptop computer, which displays the measured sensing data.

As a result of analyzing the data from all the electronic components selected for the designed WSS, it was estimated that 43, 857, 250, and 350 mW are required for power, sensing, control, and communications, respectively. Therefore, 1.5 W from the I-WPT module must be stably supplied to the WSS.

## 3. Fabrication and Measured Results

The designed WSS using the I-WPT module was installed on a test bench system consisting of a small-scale shaft and a speed controller, as shown in Figure 4. An electric motor with a speed controller was used to simulate the rotational speed. The diameter and circumference of the shaft were 200 and 628 mm, respectively. The I-WPT module, four sensors, controllers, and wireless communication modules were installed on the shaft.

Two current sensors [16], one for measuring the power input to the I-WPT Tx module and one for measuring the power delivered to the WSS, were installed on the table and the shaft, respectively, as shown in Figure 3. They were supplied with 5 V from the power converter, and were connected to the I-WPT Tx module and to the WSS. The current sensors measured a voltage drop at 0.01 Ω with a 1% sense resolution. The Hall-effect speed sensor [17] was assembled on the test bench system. A neodymium magnet was attached to the housing case of the speed controller, as shown in Figure 3. In general, the output voltage of the sensor is directly proportional to the strength of the magnetic field passing through the detector. The voltage change was perceived and calculated in revolutions per minute by the control part. The mechanical strain of the shaft was measured by detecting the change in the resistance of the strain gauge [18] in the Wheatstone bridge circuit. In this study, a quarter bridge circuit was designed, consisting of one strain gauge, two resistors, and an adjustable resistor for calibration. An amplifier [19] was added because the change in strain in the test system was too small when the shaft rotated. The amplifier was programmed with an amplifier gain of 128, corresponding to a full-scale differential input voltage of ±20 mV. The strain gauge sensor was glued to the shaft surface with an adhesive. A diode-based temperature sensor [20] with a wide temperature range (−50 to 125 °C) and a resolution of 0.1 °C was used. The temperature sensor was attached to the inner ring of the bearing to monitor bearing health.

A commercial Arduino printed circuit board (PCB) was used as the controller of the WSS attached to the shaft. To transmit and store the sensing data wirelessly to the PC, the controller board with a radio shield and a radio Tx module [21] were installed. The radio Rx module was connected to the PC. The data of the input power were sent to the PC through the controller on the table via a cable connection. 

### 3.1. Power Monitoring

For the input power, the state of power consumption according to the rotation and vibration of the shaft should be monitored in real time. The power consumption is presented in terms of the shaft speed, as shown in Figure 5. In addition, the power transmission efficiency can be estimated on the basis of the input power at the transmitting part of the I-WPT module. The input power was about 1.6 W. The power consumption of the WSS was 1.25 W, regardless of the rotational speed of the shaft. In this power monitoring, the stable power consumption characteristics, despite the rotation and vibration of the shaft, indicated that power was stably supplied from the I-WPT module, as a result of the optimized number of Tx and Rx coils in the I-WPT module [14]. The power transmission efficiency of the I-WPT module in the WSS was 78.1%.

The power consumption of the WSS largely depends upon the actual data transmission rate, sampling rate, and wireless communication distance. In this study, the transmission rate was 250 kbps, and the distance between the radio Tx and Rx module was as short as 30 cm. Moreover, most sensors have a normal sampling rate, except for strain measurement. The baud rate was set to 57.6 kbps. Thus, the power consumption of the WSS was less than the maximum power requirement.

### 3.2. Shaft Speed Monitoring

The speed sensor generates a pulse-shaped digital signal output as the shaft rotates, and the duty cycle is dependent on the shaft speed. The signal’s ON/OFF state or each duration is determined by the presence or absence of a magnetic field, as well as the speed of the shaft. The start and end times in an OFF period without a magnetic field are called Time Passed, and the number of times the sensor crosses the magnet is called the Hall count. The rotation speed of the shaft is calculated by analyzing Time Passed and Hall count. The cycling and speed calculations are handled in the program of the control part using following equations [22]:(1)$$\mathrm{Time}\;\mathrm{Passed}=\frac{t_{\mathrm{end}}-{\;t}_{\mathrm{start}}}{1,000,000}\;\;\;\;\;\left[s\right]$$
(2)$$\mathrm{RPM}\;\mathrm{value}=\frac{\mathrm{hall}\;\mathrm{count}}{\mathrm{Time}\;\mathrm{Passed}}\times 60\;\;\;\;\;\left[\mathrm{rpm}\right]$$
where t_start_ and t_end_ are times between the start and end position when the signal output is in an OFF period. 

The calculated time passed and shaft speed are presented in Figure 6a,b, respectively. These results show that the period times for one revolution at shaft speeds of 100, 200, and 300 rpm were 0.6, 0.3, and 0.2 s, respectively. The rotational speed of the shaft measured in the WSS was verified using a tachometer, as shown in Figure 6c. Therefore, it was confirmed that the speed sensor of the WSS accurately measured the shaft speed. 

### 3.3. Strain Monitoring

The voltage source for excitation to power the Wheatstone bridge and the sensing output voltage were measured, in order for the correct strain to be calculated with equations [23]. The vibration measurement system for the propulsion shaft, presented in Figure 4, offers wireless transmission of signals from the strain gauge on the rotating shaft. The strain maxima were recorded under ±45° with reference to the shaft axis. Figure 7 shows the strain characteristics versus time for the experimental test, without a load, under different rotating speeds. As can be seen, the vibration changed with the rotational speed, whereas the amplitude of the strain became larger with increasing rotational speed. The strain fluctuated from −0.003 to 0.006, −0.0032 to 0.0035, and −0.0087 to 0.0006 at three experimental speeds of 100, 200, and 300 rpm, respectively. Under no load conditions of the shaft, the strain value was very low.

### 3.4. Temperature Monitoring

The temperature sensor generated its output as an analog voltage signal, and this output signal was converted into temperature by the program of the controller. Figure 8 shows the measured temperature results of the bearings at shaft speeds of 100, 200, and 300 rpm. As can be seen, the measured temperatures were nearly identical at all speeds. This meant that the bearings operated normally, without any unusual vibrations around them. In general, frictional heat is generated due to abnormal operation of the shaft system or the bearings, and the temperature around the bearings rises.

## 4. Conclusions

We designed and installed an entire WSS (without a battery) on a rotating shaft for real-time monitoring of its status. The WSS consisted of four sensors for current, speed, vibration, and temperature; the controller module, the radio module, and the I-WPT module. Firstly, to provide the required power of 1.75 W to drive the WSS on the rotating shaft, an I-WPT system using multiple coils was implemented. Secondly, the whole WSS integrating four sensors on the rotary shaft could be designed, due to the sufficient power transfer performance of the implemented I-WPT module. Finally, the whole WSS was installed on a small-scaled test bench system, and a demonstration of the monitoring of the propulsion shaft’s status in real time without a battery was presented. For power monitoring with the current sensor, 1.6 W was transmitted wirelessly to the WSS, and 1.25 W was consumed to drive the WSS. It was observed that power was stably supplied and consumed, regardless of the rotational speed of the shaft. The rotational speed of the shaft was measured with the speed sensor, and its accuracy was verified using a tachometer. Vibration and frictional heat, which can occur due to abnormal shaft operation, were measured using a strain gauge sensor and a temperature sensor, respectively. In these experiments, all data generated by all sensors of the WSS were wirelessly transmitted to the PC through the radio module for storage and analysis. Therefore, the proposed and developed WSS proved that the state of the propulsion shaft could be measured in real time without requiring battery power. 

## Figures and Tables

**Figure 1 sensors-23-00558-f001:**
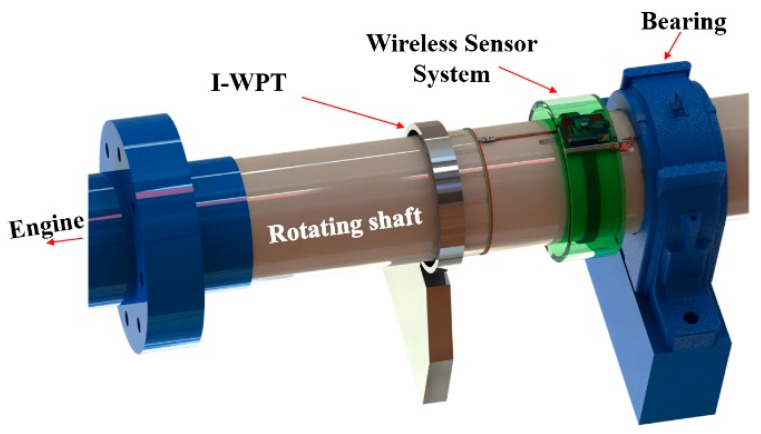
Installation of the WSS using the I-WPT unit on the rotating shaft.

**Figure 2 sensors-23-00558-f002:**
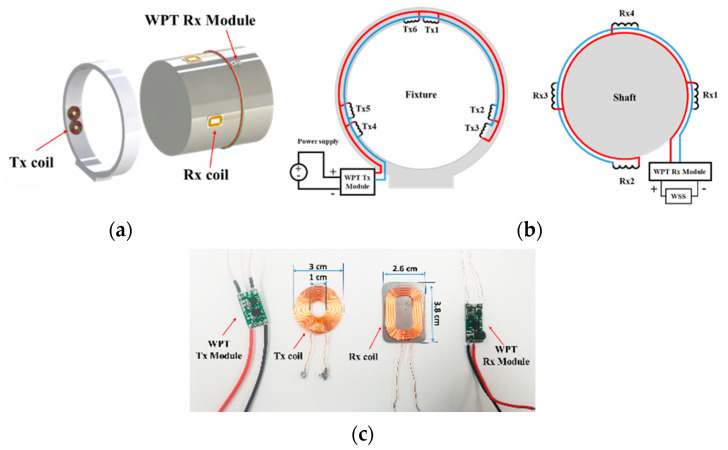
A 3-D design of the I-WPT module (**a**), equivalent circuits of the I-WPT module (**b**), and photos of the Tx and Rx coils and each module (**c**).

**Figure 3 sensors-23-00558-f003:**
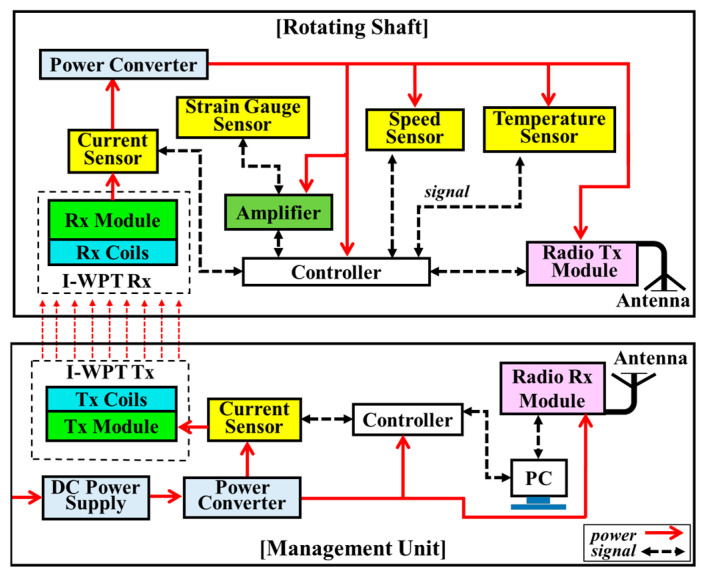
Block diagram of the entire WSS, integrating four sensors on the rotating shaft.

**Figure 4 sensors-23-00558-f004:**
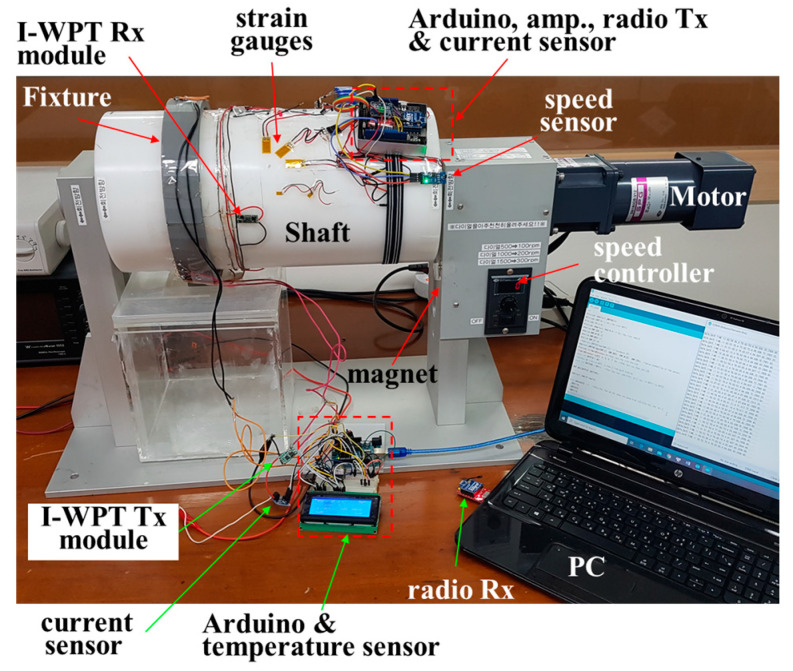
The WSS installed on the test bench system.

**Figure 5 sensors-23-00558-f005:**
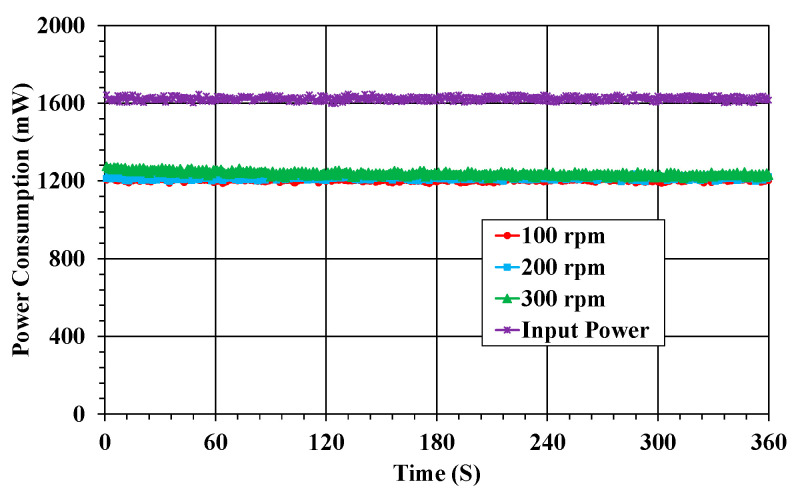
Power monitoring results in the WSS.

**Figure 6 sensors-23-00558-f006:**
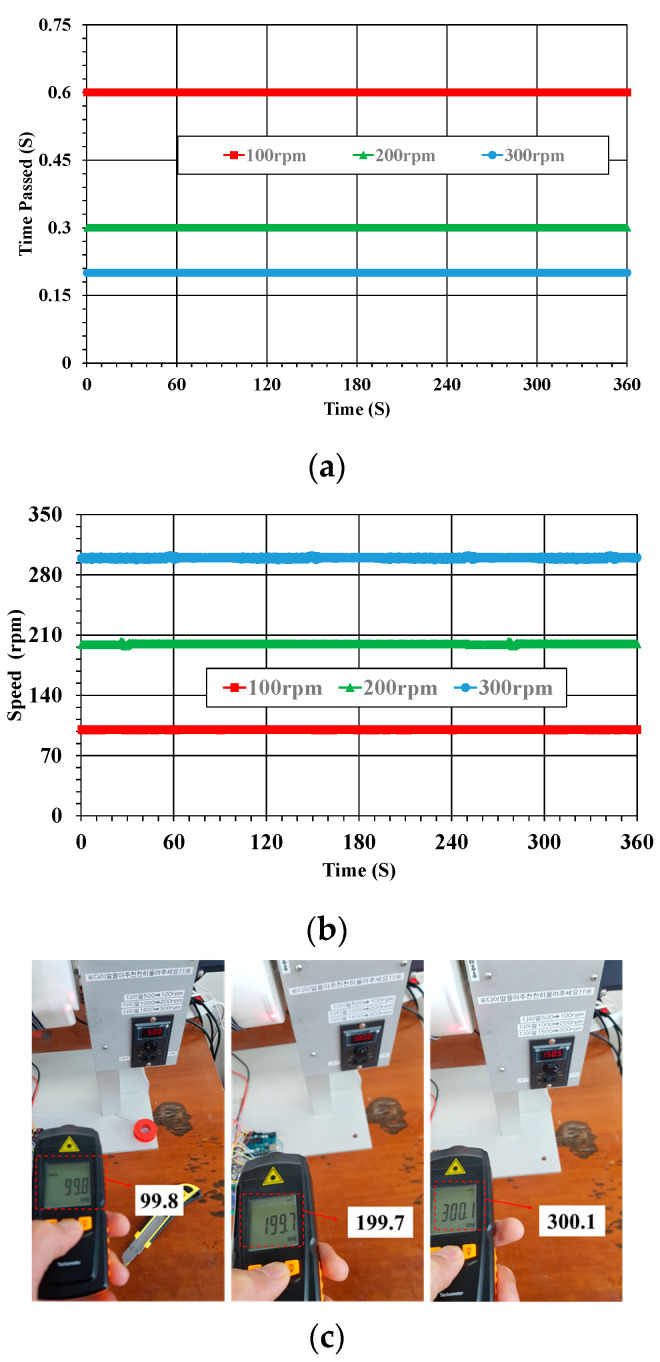
Calculated Time Passed (**a**), the shaft speed (**b**), and tachometer test results (**c**).

**Figure 7 sensors-23-00558-f007:**
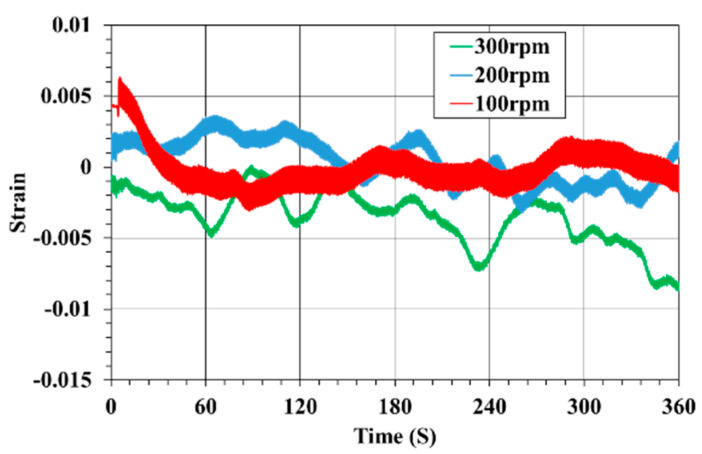
Experimental results of vibration measurements at a relative orientation of 45°.

**Figure 8 sensors-23-00558-f008:**
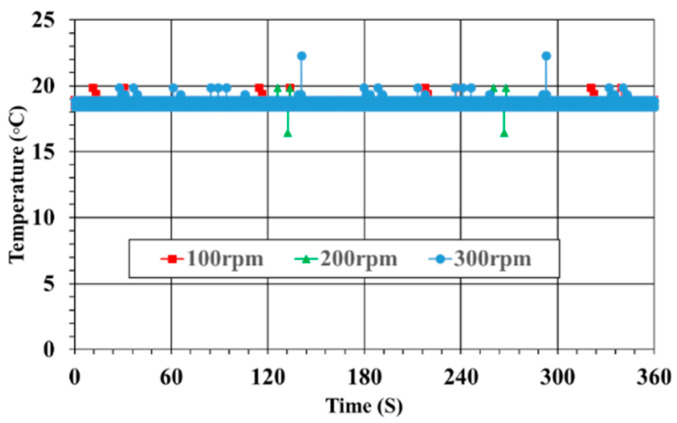
Bearing temperatures.

## Data Availability

The datasets used during the current study are available from the corresponding author on reasonable request.

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
