# Peer review of "Battery-Free and Real-Time Wireless Sensor System on Marine Propulsion Shaft Using a Wireless Power Transfer Module"

_sensors, 2023, doi:10.3390/s23020558_

Round 1

Reviewer 1 Report

While I believe the use case addressed by this research is relevant, as far as the manuscript is concerned, I believe that more details about the system would give the reader a more complete view. I believe that, even if the system has already been discussed in [16], it would be necessary and appropriate to reiterate the essential elements and components of the system. 

Therefore, I would invite the authors to report in this manuscript more details about the type of connectivity used, Bluetooth, Zigbee, the sensors used, are they custom or commercially available?. Finally, even a more detailed photograph of the two boards present at the primary and secondary would give greater value to this manuscript.

Author Response

Dear Reviewer,

Thanks for the good comments.
According to your opinions, the abstract and introduction have been modified and details of the I-WPT was added to the main body 2-1.
Please review and comment again.

Reviewer 2 Report

The paper presented the development and testing of a real-time wireless sensor system for a marine propulsion shaft using the concept of wireless power transfer.

The developed system consists of four sensors and two modules. The four sensors, the current, speed, strain gauge, and temperature sensor were used for power monitoring, performance monitoring, mechanical strain on the shaft, and bearing health, respectively. The control module, implemented using an Arduino microprocessor, was responsible for data processing, data display, and control of sensors and shaft modules. On the other hand, the communication module transmits data to a PC using a transceiver system. The developed system was implemented on a test bench, and the results were presented. The obtained results are exciting and relevant.

Technical comments

The authors should explicitly highlight the contributions of the paper. It is difficult to identify the key contributions in the current writeup.

 There is a need to discuss the results concerning what is obtainable using existing techniques and equipment. This will give a basis for the comparison of results with existing results.

Author Response

(The authors gave the same response as above.)

Reviewer 3 Report

The title does not match most of the manuscript's contents. Are you presenting a sensor strategy, method, or analysis? The article title must provide a succinct description of the article's content. Each word is carefully chosen to convey the most information in the manuscript.

The abstract should answer these questions:

- What was done?

- Why are you doing it?

- What did you establish as a result of your research?

- Why is this data useful and important?

Numerical results obtained should be added to the abstract.

What is the contribution of the study?. it’s not cleared.

In the section Introduction, it is necessary to justify the relevance to the current scientific topics. I pay attention - not to the relevance of this work (article), but exactly the topic. It is necessary to give arguments in favor of the fact that it is very important to carry out research on this topic, and that the results of such research are needed in practice.

After all, it may turn out that the topic itself is no longer needed, outdated because science is already much further ahead. In this case, why waste time on research? Note that relevance is understood precisely in the sense of the importance of this scientific topic (issue), not in the sense of this work (article).

If such arguments will be given, it is clear why further analysis of the literature - since this scientific topic is essential, it is necessary to understand what achievements in the research of this topic already have and what do not, and therefore requires a new study.

Thus, the logic of the construction of the article should be as follows:

The Introduction section proves that this research topic needs to be dealt with, hence the Literature review and problem statement section needs to identify what parts of the problem are unresolved and require research, hence the justified purpose of the research which is set out in The aims and objectives of the research section.

The discussion of the results is preferred to be like this:

Answer the question, what explains the results obtained? When answering this question, it is necessary to refer to those objects in the article, which display the discussed result. Such objects are formulas, figures, and tables.

What are the peculiarities of the proposed method and the obtained results in comparison with the existing ones (it is necessary to make a comparison with the known data, indicating the references to the relevant works of colleagues)

All the presented results except Figure 6 are of constant measurements. Therefore, either you add dynamic variation or show them using a table.

For Figure 5 (b), when you show a graphical result, you need to show some dynamic change such as showing one dynamic speed transient with stepping up (100 to 200 to 300). The same thing is also valid in Figure 5 (a).

How you verified your results? For example, in Figure 6, you need to cite a reference or a model equation that agrees with your measurements.

Author Response

Dear Reviewer,
The authors would like to thank the reviewer for their valuable and helpful comments. The manuscript has been revised based on the reviewers' comments. The authors tried to address all of the reviewers' comments in the best way possible. Detailed responses are provided in the following. Please note that the reviewer's comments are in blue, while our answers are in black in this response letter.

Regards

Lee

Round 2

Reviewer 1 Report

The paper is as good as it is. No further comments from my side.

Author Response

Dear Reviewer,

The authors would like to thank the reviewer for their valuable and helpful comments. The manuscript has been revised based on other reviewers' comments.

Reviewer 2 Report

There is a need to improve the flow of the manuscript. The introduction has some paragraphs that are just two sentences long. Kindly address.

The paper's contributions in relation to existing works are still unclear. I recommend that the authors enumerate and justify the significant contributions of the paper, preferably in a list.

Author Response

REVIEWER 2

There is a need to improve the flow of the manuscript. The introduction has some paragraphs that are just two sentences long. Kindly address.

 The paper's contributions in relation to existing works are still unclear. I recommend that the authors enumerate and justify the significant contributions of the paper, preferably in a list.

  • Authors appreciate the comments of the reviewer. The contributions were classified into three categories and explained respectively. We have revised Abstract, Introduction, Body, and Conclusion to reflect your comments. Please check the marked parts in the revised manuscript.

Reviewer 3 Report

The author couldn't convince the reviewer when justifying the previous last three issues under discussion, which are as follows:

All the presented results except Figure 6 are of constant measurements. Therefore, either you add dynamic

variation or show them using a table.

For Figure 5 (b), when you show a graphical result, you need to show some dynamic change such as showing one

dynamic speed transient with stepping up (100 to 200 to 300). The same thing is also valid in Figure 5 (a).

How you verified your results? For example, in Figure 6, you need to cite a reference or a model equation that agrees with your measurements.

Please answer the above comments one by one.

Author Response

REVIEWER 3

The author couldn't convince the reviewer when justifying the previous last three issues under discussion, which are as follows:

All the presented results except Figure 6 are of constant measurements. Therefore, either you add dynamic variation or show them using a table.

For Figure 5 (b), when you show a graphical result, you need to show some dynamic change such as showing one dynamic speed transient with stepping up (100 to 200 to 300). The same thing is also valid in Figure 5 (a).

- In our previous work [14], we carried out optimization of the proposed I-WPT module as shown in the following figures [14]. In this work, the I-WPT having the optimized number of Tx and Rx coils showed constant voltage and power characteristics regardless shaft speed. If the shaft is connected to the magnetic flux without interruption as it rotates, the power depends on the number of turn of the coils. So, in this work, the number of Tx and Rx coils and the number of its turn of each coil were fixed after optimization process.

How you verified your results? For example, in Figure 6, you need to cite a reference or a model equation that agrees with your measurements.

Please answer the above comments one by one.

  • The author appreciates the comments of the reviewers. We have revised section 3.2 Shaft Speed Monitoring adding some equations [22].
